# ONE-SHOT NEURAL ARCHITECTURE SEARCH VIA COMPRESSIVE SENSING

## ABSTRACT

Neural architecture search (NAS), or automated design of neural network models, remains a very challenging meta-learning problem. Several recent works (called "one-shot" approaches) have focused on dramatically reducing NAS running time by leveraging proxy models that still provide architectures with competitive performance. In our work, we propose a new meta-learning algorithm that we call CoNAS, or Compressive sensing-based Neural Architecture Search. Our approach merges ideas from one-shot NAS approaches with iterative techniques for learning low-degree sparse Boolean polynomial functions. We validate our approach on several standard test datasets, discover novel architectures hitherto unreported, and achieve competitive (or better) results in both performance and search time compared to existing NAS approaches. Further, we provide theoretical analysis via upper bounds on the number of validation error measurements needed to perform reliable meta-learning; to our knowledge, these analysis tools are novel to the NAS literature and may be of independent interest.

## 1 INTRODUCTION

**Motivation.** Choosing a suitable neural network architecture for complex prediction tasks such as image classification and language modeling often requires a substantial effort of trial-and-error. Therefore, there has been a growing interest to *automatically learn* (or meta-learn) the architecture of neural networks that can achieve competitive (or better) results over hand-designed architectures. The sub-field of neural architecture search (NAS) addresses the problem of designing competitive architectures with as small computational budget as possible.

Numerous approaches for neural architecture search already exist in the literature, each with their own pros and cons: these include black-box optimization based on reinforcement learning (RL) (Zoph & Le, 2017), evolutionary search (Real et al., 2019), and Bayesian optimization (Cao et al., 2019; Kandasamy et al., 2018). Though the algorithmic details vary, most of these NAS methods face the common challenge of evaluating the test/validation performance of a (combinatorially) large number of candidate architecture evaluations.

**Our Contributions.** In a departure from traditional methods, we approach the NAS problem via the lens of *compressive sensing*. The field of compressive sensing (or sparse recovery), introduced by the seminal works of (Candes et al., 2006; Donoho et al., 2006), has received significant attention in both ML theory and applications over the last decade, and has influenced the development of numerous advances in nonlinear and combinatorial optimization.

We leverage these advances for the NAS problem. In particular, we develop a new NAS method called CoNAS (Compressive sensing-based Neural Architecture Search), which merges ideas from sparse recovery with so-called "one-shot" architecture search methods (Bender et al., 2018), described in greater detail below. CoNAS consists of two new innovations: (i) a new *search space* that permits exploration of a large(r) number of diverse candidate architectures, and (ii) a new *search strategy* that borrows ideas from recovery of Boolean functions from their (sparse) Fourier expansions.

Our experiments show that CoNAS is able to discover a deep convolutional neural network with test error $2.74 \pm 0.12\%$ on CIFAR-10 classification, outperforming existing state-of-the-art methods, including DARTs (Liu et al., 2018b), ENAS (Pham et al., 2018), random search with weight-sharing (RSWS) (Li & Talwalkar, 2019), and the baseline vanilla random search method (Liu et al., 2018b) in

terms of test error, search time, model size, and number of multiply-add operations. Moreover, CoNAS can achieve the comparable performance as NASNet (Zoph et al., 2018) and AmoebaNet (Real et al., 2019) with less than one GPU-day of computation. Our experiments on designing recurrent neural networks for language modeling are somewhat short of the state-of-the-art (Zilly et al., 2017), but we find that CoNAS still finds competitive results with less search time than previous NAS approaches. Our results are exactly reproducible (having been trained with fixed pseudorandom seeds), and an implementation of CoNAS will be made publicly available post-peer review.

Finally, while our original motivation was to devise an empirically useful NAS method, a nice benefit is that CoNAS can also be *theoretically analyzed*, since existing theoretical results for Fourier-sparse Boolean functions can be ported over in order to provide upper bounds for the required number of performance evaluations of sub-architectures of the one-shot model. This, to our knowledge, is one of the first results of their kind in the NAS literature and may be of independent interest. We defer discussion of our approach to Section 3.

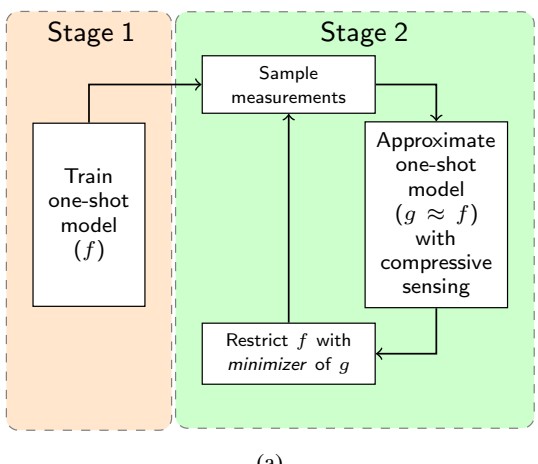

(a)

Figure 1: *Overview of CoNAS. A one-shot neural network model $f$ is pre-trained, and an appropriate sub-graph of $f$ is chosen by applying a sparse recovery technique. Iterative sparse recoveries allow to find the larger sub-graph from $f$.*

**Our Techniques.** The intuition behind compressive sensing is that if a signal (or function) can be represented via a sparse basis expansion, then it can be recovered (either exactly or approximately) from a small number of randomized measurements. CoNAS leverages this intuition in the context of one-shot architecture search (Bender et al., 2018). In one-shot NAS, instead of evaluating several candidate architectures, a single "base" neural network model is pre-trained; a class of sub-networks is identified (called the search space) and the performance of each sub-network is evaluated on a validation set; and the best-performing sub-network is finally selected and fine-tuned.

Following Hazan et al. (2017); Stobbe & Krause (2012), we model the sub-network selection as a sparse recovery problem. Concretely, consider a function $f$ that maps sub-architectures to a measure of performance (validation loss). We assume that $f$ can be written as a sparse, low-degree polynomial in the (discrete) Fourier basis . If the sparsity assumption is satisfied, then we claim the function $f$ can be reconstructed using a very small number of sub-network evaluations, thus reducing overall compute time. A key challenge lies in defining a suitable search space; we propose one that is considerably larger than the one used in DARTS or ENAS, allowing us to (putatively) search over a more diverse set of candidate architectures.

## 2 BACKGROUND

We briefly describe one-shot neural architecture search techniques (Bender et al., 2018; Li & Talwalkar, 2019); a supplementary description is available in Appendix A.1. Following the treatment given in the recent survey paper (Elsken et al., 2018), one-shot NAS approaches have three main components: a search space, a search strategy, and a performance estimation strategy.

**Search Space.** The goal of one-shot NAS is to find the best performing *cell*, a fundamental component from which more complex architectures are constructed via stacking. Following Liu et al. (2018b), a cell is a directed acyclic graph (DAG) where a node corresponds to the latent representation, and a directed edge transforms predecessor nodes using a given *operation*; common operations used in CNNs include $3 \times 3$ and $5 \times 5$ separable convolutions, $3 \times 3$ max pooling, and $3 \times 3$ average pooling. Each cell has two input nodes and one output node, and intermediate nodes

can only be connected by predecessor nodes including input nodes. Intermediate nodes are wired to two predecessor nodes in CNNs and one predecessor node in RNNs.

**Search Strategy and Performance Estimation Strategy.** Having defined a search space, one-shot NAS approaches employ four steps: (i) train a single "one-shot" base model that is capable of predicting the performance of sub-architectures[1]; (ii) *randomly sample* sub-architectures of a trained one-shot model and measure performance over a hold-out validation set of samples; (iii) select the candidate (cell) with best validation performance; (iv) retrain a deeper final architecture using the best cell. Using the one-shot model as the proxy measurements of the candidate architecture corresponds to the performance estimation strategy (first and second step).

**Fourier analysis of Boolean functions.** We follow the treatment given in O'Donnell (2014). A real-valued Boolean function is one that maps $n$-bit binary vectors (i.e., nodes of the hypercube) to real values: $f : \{-1, 1\}^n \to \mathbb{R}$. Such functions can be represented in a basis comprising real multilinear polynomials called the *Fourier* basis, defined as follows. (We denote the vectors with bold letters. Also, $[n]$ denotes the set $\{1, 2 \ldots, n\}$.)

**Definition 2.1.** *For $S \subseteq [n]$, define the parity function $\chi_S : \{-1, 1\}^n \to \{-1, 1\}$ such that $\chi_S(\boldsymbol{\alpha}) = \prod_{i \in S} \alpha_i$. Then, the Fourier basis is the set of all $2^n$ parity functions $\{\chi_S\}$.*

The key fact is that the basis of parity functions forms an $K$-bounded orthonormal system (BOS) with $K = 1$, therefore satisfying two properties:

$$\langle \chi_S, \chi_T \rangle = \begin{cases} 1, & \text{if } S = T \\ 0, & \text{if } S \neq T \end{cases} \quad \text{and} \quad \sup_{\boldsymbol{\alpha} \in \{-1,1\}^n} |\chi_S(\boldsymbol{\alpha})| \leq 1 \text{ for all } S \subseteq [n], \quad (2.1)$$

Due to orthonormality, any Boolean function $f$ has a unique Fourier representation, given by $f(\boldsymbol{\alpha}) = \sum_{S \subseteq [n]} \hat{f}(S) \chi_S(\boldsymbol{\alpha})$, with Fourier coefficients $\hat{f}(S) = \mathbb{E}_{\boldsymbol{\alpha} \in \{-1,1\}^n}[f(\boldsymbol{\alpha})\chi_S(\boldsymbol{\alpha})]$ where expectation is taken with respect to the uniform distribution over the nodes of the hypercube.

A modeling assumption is that the Fourier spectrum of the function is concentrated on monomials of small degree ($\leq d$). This corresponds to the case where $f$ is a decision tree (Hazan et al., 2017), and allows us to simplify the Fourier expansion by limiting its support. Let $\mathcal{P}_d \subseteq 2^{[n]}$ be a fixed collection of Fourier basis such that $\mathcal{P}_d := \{\chi_S \subseteq 2^{[n]} : |S| \leq d\}$. Then $\mathcal{P}_d \subseteq 2^{[n]}$ induces a function space consisting of all functions of order $d$ or less, denoted by $\mathcal{H}_{\mathcal{P}_d} := \{f : Supp[\hat{f}] \subseteq \mathcal{P}_d\}$. For example, $\mathcal{P}_2$ allows us to express the function $f$ with at most $\sum_{l=0}^d \binom{n}{l} \equiv \mathcal{O}(n^2)$ Fourier coefficients.

Lastly, if we have prior knowledge of some set of bits $J$, we use an operation called *restriction*.

**Definition 2.2.** *Let $f : \{-1, 1\}^n \to \mathbb{R}$, $(J, \overline{J})$ be a partition of $[n]$, and $z \in \{-1, 1\}^{\overline{J}}$. The restriction of $f$ to $J$ using $z$ denoted by $f_{J|z} : \{-1, 1\}^J \to \mathbb{R}$ is the subfunction of $f$ given by fixing the coordinates in $\overline{J}$ to the bit values $z$.*

## 3 PROPOSED ALGORITHM: CoNAS

**Overview.** Our proposed algorithm, Compressive sensing-based Neural Architecture Search (CoNAS), infuses ideas from learning a sparse graph (Boolean Fourier analysis) into one-shot NAS. CoNAS consists of two novel components: an expanded search space, and a more effective search strategy.

**Search Space.** Our first ingredient is an expanded search space. Following the approach of DARTS (Liu et al., 2018b), we define a directed acyclic graph (DAG) where all predecessor nodes are connected to every intermediate node with all possible operations. We represent any sub-graph of the DAG using a binary string $\boldsymbol{\alpha}$ called the *architecture encoder*. Its length is the total number of edges in the DAG, and a 1 (resp. $-1$) in $\boldsymbol{\alpha}$ indicates an active (resp. inactive) edge.

---

[1]We note that the quality of sub-network performance predictions is heavily dependent on the base model that is trained. Choosing the correct base model is itself a separate challenge, which earlier papers such as Bender et al. (2018) have addressed in detail. We do not pursue that direction here since our focus is on the sub-network selection problem, and assume that the base model is well-trained.

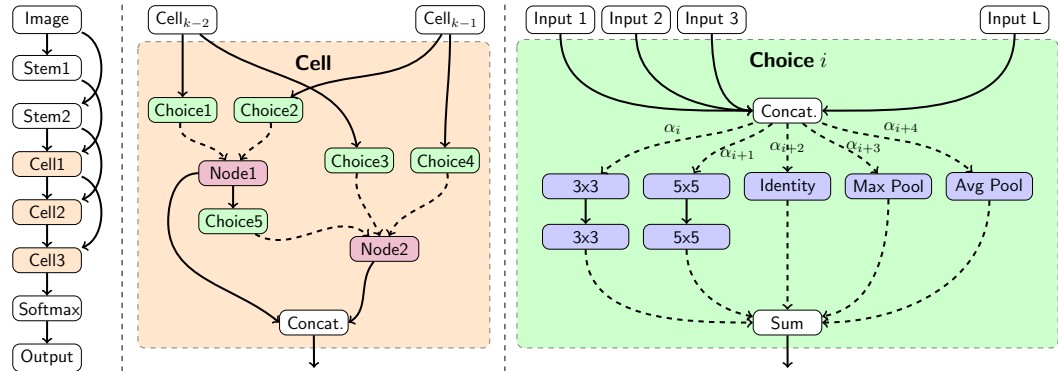

Figure 2: *Diagram inspired by* Bender et al. (2018). *The example architecture encoder $\boldsymbol{\alpha}$ samples the sub-architecture for $N = 5$ nodes (two intermediate nodes) with five different operations. Each component in $\boldsymbol{\alpha}$ maps to the edges one-to-one in all* Choice *blocks in a cell. If a bit in $\boldsymbol{\alpha}$ corresponds to $1$, the edge activates, while $-1$ turns off the edge. Since the CNN search space finds both* normal cell *and* reduce cell, *the length of $\boldsymbol{\alpha}$ is equivalent to $(2 + 3) \cdot 5 \cdot 2 = 50$.*

Figure 2 gives an example of how the architecture encoder $\boldsymbol{\alpha}$ samples the sub-architecture of the fully-connected model in case of a convolutional neural network. The goal of CoNAS is to find the "best" encoder $\boldsymbol{\alpha}^*$, which is "close enough" to the global optimum returning the best validation accuracy by constructing the final model with $\boldsymbol{\alpha}^*$ encoded sub-graph.

Since each edge can be switched on and off independently, the proposed search space allows exploring a cell with more diverse connectivity patterns than DARTS Liu et al. (2018b). Moreover, the number of possible configurations exceeds similar previously proposed search spaces with constrained wiring rules Li & Talwalkar (2019); Pham et al. (2018); Real et al. (2019); Zoph et al. (2018).

**Search Strategy.** We propose a compressive measuring strategy to approximate the one-shot model with a Fourier-sparse Boolean function. Let $f : \{-1, 1\}^n \to \mathbb{R}$ map the sub-graph of the one-shot pre-trained model encoded by $\boldsymbol{\alpha}$ to its validation performance. Similar to Hazan et al. (2017), we collect a small number of function evaluations of $f$, and reconstruct the Fourier-sparse function $g \approx f$ via sparse recovery algorithms with randomly sampled measurements. Then, we solve $\arg\min_{\boldsymbol{\alpha}} g(\boldsymbol{\alpha})$ by exhaustive enumeration over all coordinates in its sub-cube $\{-1, 1\}^{\overline{J}}$ where $(J, \overline{J})$ partitions $[n]$ (Definition 2.2)[2]. If the solution of the $\arg\min_{\boldsymbol{\alpha}} g(\boldsymbol{\alpha})$ does not return enough edges to construct the cell (some intermediate nodes are disconnected), we simply connect the intermediate nodes to the previous cell output, Cell$_{k-2}$, using the Identity operation (this does not increase neither the model size nor number of multiply-add operations). Larger cells can be found from multiple iterations by restricting the approximate function $g$ and with fixing the bit values found in the previous solution, and randomly sampling sub-graphs in the remaining edges.

**Full Algorithm.** We now describe CoNAS in detail, with pseudocode provided in Appendix A.3. We first train a one-shot model with standard backpropagation but only updates the weight corresponding to the randomly sampled sub-graph edges for each minibatch. Then, we randomly sample sub-graphs by generating architecture encoder strings $\boldsymbol{\alpha} \in \{-1, 1\}^n$ using a $Bernoulli(p)$ distribution for each bit of $\boldsymbol{\alpha}$ independently (We set $p = 0.5$).

In the second stage, we collect $m$ measurements of randomly sampled sub-architecture performance denoted by $\mathbf{y} = (f(\boldsymbol{\alpha}_1), f(\boldsymbol{\alpha}_2), \ldots, f(\boldsymbol{\alpha}_m))^T$. Next, we construct the *graph-sampling matrix* $\mathbf{A} \in \{-1, 1\}^{m \times |\mathcal{P}_d|}$ with entries

$$\mathbf{A}_{l,k} = \chi_{S_k}(\boldsymbol{\alpha}_l), \qquad l \in [m], k \in [|\mathcal{P}_d|], S \subseteq [n], |S| \le d, \tag{3.1}$$

where $d$ is the maximum degree of monomials in the Fourier expansion, and $S_k$ is the index set corresponding to k[th] Fourier basis.

---

[2]This is similar to the idea of de-biasing in the *Hard-Thresholding (HT)* algorithm (Foucart & Rauhut, 2017) where the support is first estimated, and then within the estimated support, the coefficients are calculated through least-squares estimation.

We solve the familiar Lasso problem (Tibshirani, 1996):

$$\mathbf{x}^* = \underset{\mathbf{x} \in \mathbb{R}^{|\mathcal{P}_d|}}{\arg \min} \|\mathbf{y} - \mathbf{A}\mathbf{x}\|_2^2 + \lambda \|\mathbf{x}\|_1, \tag{3.2}$$

to (approximately) recover the global optimizer $\mathbf{x}^*$, the vector contains the Fourier coefficients corresponding to $\mathcal{P}_d$. We define an approximate function $g \approx f$ with Fourier coefficients with the top-$s$ (absolutely) largest coefficients from $\mathbf{x}^*$, and compute $\boldsymbol{\alpha}^* = \arg \min_{\boldsymbol{\alpha}} g(\boldsymbol{\alpha})$, resulting all the possible points in the subcube defined by the support of $g$ (this computation is feasible if $s$ is small). Multiple stages of sparse recovery (with successive restrictions to previously obtained optimal $\boldsymbol{\alpha}^*$) enable us to approximate additional monomial terms. Finally, we obtain a cell to construct the final architecture by activating the edges corresponding to all $i \in [n]$ such that $\alpha_i^* = 1$.

**Theoretical support for CoNAS.** The system of linear equations $\mathbf{y} = \mathbf{A}\mathbf{x}$ with the graph-sampling matrix $\mathbf{A} \in \{-1, 1\}^{m \times O(n^d)}$, measurements $\mathbf{y} \in \mathbb{R}^m$, and Fourier coefficient vector $\mathbf{x} \in \mathbb{R}^{O(n^d)}$ is an ill-posed problem when $m \ll O(n^d)$ for large $n$. However, if the graph-sampling matrix satisfies *Restricted Isometry Property (RIP)*, the sparse coefficients, $\mathbf{u}$ can be recovered:

**Definition 3.1.** *A matrix $\mathbf{A} \in \mathbb{R}^{m \times \mathcal{O}(n^d)}$ satisfies the restricted isometry property of order $s$ with some constant $\delta$ if for every $s$-sparse vector $\mathbf{u} \in \mathbb{R}^{\mathcal{O}(n^d)}$ (i.e., only $s$ entries are non-zero) the following holds:*

$$(1 - \delta)\|\mathbf{u}\|_2^2 \leq \|\mathbf{A}\mathbf{u}\|_2^2 \leq (1 + \delta)\|\mathbf{u}\|_2^2.$$

We defer the history of improvements on the upper bounds of the number of rows from bounded orthonormal dictionaries (matrix $\mathbf{A}$) for which $\mathbf{A}$ is guaranteed to satisfy the restricted isometry property with high probability in Appendix A.2. To the best of our knowledge, the best known result with mild dependency on $\delta$ (i.e., $\delta^{-2}$) is due to Haviv & Regev (2017), which we can apply for our setup. It is easy to check that the graph-sampling matrix $\mathbf{A}$ in our proposed CoNAS algorithm satisfies BOS for $K = 1$ (Eq 3.1).

**Theorem 3.2.** *Let the graph-sampling matrix $A \in \{-1, 1\}^{m \times \mathcal{O}(n^d)}$ be constructed by taking $m$ rows (random sampling points) uniformly and independently from the rows of a square matrix $\mathbf{M} \in \{-1, 1\}^{\mathcal{O}(n^d) \times \mathcal{O}(n^d)}$. Then the normalized matrix $\mathbf{A}$ with $m = \mathcal{O}(\log^2(\frac{1}{\delta})\delta^{-2} s \log^2(\frac{s}{\delta}) d \log(n))$ with probability at least $1 - 2^{-\Omega(d \log n \log(\frac{s}{\delta}))}$ satisfies the restricted isometry property of order $s$ with constant $\delta$; as a result, every $s$-sparse vector $\mathbf{u} \in \mathbb{R}^{\mathcal{O}(n^d)}$ can be recovered from the sample $y_i$'s:*

$$\mathbf{y} = \mathbf{A}\mathbf{u} = \Big( \sum_{j=1}^{|\mathcal{O}(n^d)|} u_j \mathbf{A}_{i,j} \Big)_{i=1}^m,$$

*by LASSO (equation 3.2).*

*Proof.* First, we note that the graph-sampling matrix $A$ is a BOS matrix with $K = 1$; hence, directly invoking Theorem 4.5 of Haviv & Regev (2016) to our setting, we can see that matrix $A$ satisfies RIP. Now according to Theorem 1.1 of Candes (2008), letting $\delta < \sqrt{2} - 1$, the $l_1$ minimization or LASSO will recover exactly the $s$ sparse vector $u$. For instance, in our experiments, we have selected $m = 1000$ which is consistent with our parameters, $d = 2, s = 10, n = 140$. $\qquad\square$

Here, it is worthwhile to mention two points: first, the above upper bound on the number of rows of the graph-sampling matrix $A$ is the tightest bound (according to our knowledge) for the BOS matrices to satisfy RIP. There exist series of results establishing the RIP for BOS matrices during the last 15 year. We have reviewed these results in the appendix A.2. Second, instead of LASSO, one can use any sparse recovery method (such as IHT (Blumensath & Davies, 2009)) in our algorithm. In essence, Theorem 3.2 provides a successful guarantee for recovering the optimal sub-network of a given size given a sufficient number of performance measurements.

## 4  EXPERIMENTS AND RESULTS

We experiment on two different NAS problems: (i) CNN search on CIFAR-10, CIFAR-100, Fashion MNIST and SVHN, (ii) an RNN search for Penn Treebank (PTB). We describe the training details

Table 1: ***Comparison with hand-designed networks and state-of-the-art NAS methods on CIFAR-10** (Lower test error is better). The results are grouped as follows: manually designed networks, published NAS algorithms, and our experimental results. The average test error of our experiment used five random seeds. Table entries with "-" indicates that either the field is not applicable or unknown. The methods listed in this table are trained with auxiliary towers and cutout augmentation. Running time cost is measured on NVIDIA TITAN X GPU. The reported time of CoNAS includes both training one-shot model and gathering measurements for the sparse recovery.*

| Architecture | Test Error (%) | Params (M) | Multi-Add (M) | Search GPU days |
|---|---|---|---|---|
| PyramidNet (Yamada et al., 2018) | 2.31 | 26 | - | - |
| AutoAugment (Cubuk et al., 2019) | 1.48 | 26 | - | - |
| ProxylessNAS (Cai et al., 2019) | 2.08 | 5.7 | - | 4 |
| NASNet-A (Zoph et al., 2018) | 2.65 | 3.3 | - | 2000 |
| AmoebaNet-B (Real et al., 2019) | $2.55 \pm 0.05$ | 2.8 | - | 3150 |
| GHN[+] (Zhang et al., 2018) | $2.84 \pm 0.07$ | 5.7 | - | 0.84 |
| SNAS (Xie et al., 2018) | $2.85 \pm 0.02$ | 2.8 | - | 1.5 |
| ENAS (Pham et al., 2018) | 2.89 | 4.6 | - | 0.45 |
| DARTs (Liu et al., 2018b) | $2.76 \pm 0.09$ | 3.3 | 548 | 4 |
| Random Search (Liu et al., 2018b) | $3.29 \pm 0.15$ | 3.1 | - | 4 |
| ASHA (Li & Talwalkar, 2019) | $2.85 \| 3.03 \pm 0.13$ | 2.2 | - | - |
| RSWS (Li & Talwalkar, 2019) | $2.71 \| 2.85 \pm 0.08$ | 3.7 | 634 | 2.7 |
| DARTs[#] (Li & Talwalkar, 2019) | $2.62 \| 2.78 \pm 0.12$ | 3.3 | - | 4 |
| DARTs[†] | $2.59 \| 2.78 \pm 0.13$ | 3.4 | 576 | 4 |
| **CoNAS (t=1)** | **2.57**$\| 2.74 \pm 0.12$ | **2.3** | **386** | **0.4** |
| **CoNAS (t=4)** | $2.55 \| 2.62 \pm 0.06$ | 4.8 | 825 | 0.5 |
| **CoNAS (t=1, C=60)[+]+AutoAugment** | 1.87 | 6.1 | 1019 | 0.4 |

[#] DARTS experimental results from Li & Talwalkar (2019).
[†] Used DARTS search space with five operations for direct comparisons.
[+] 'C' stands for the number of initial channels. Trained 1000 epochs with AutoAugment.

for CIFAR-10 and PTB in Sections 4.1 and 4.3 respectively. Our evaluation setup for training the final architecture (CIFAR-10 and PTB) is the same as that reported in DARTS and RSWS.

## 4.1 CIFAR-10

**Architecture Search.** We create a one-shot architecture similar to RSWS with a cell containing $N = 7$ nodes with two nodes as input and one node as output; our wiring rules between nodes are different and as in Section 3. We used five operations: $3 \times 3$ and $5 \times 5$ separable convolutions, $3 \times 3$ max pooling, $3 \times 3$ average pooling, and Identity. On CIFAR-10, we equally divide the 50,000-sample training set to training and validation sets, following Li & Talwalkar (2019) and Liu et al. (2018b). We train a one-shot model by sampling the random sub-graph under Bernoulli(0.5) sampling with eight layers and 16 initial channels for 100 training epochs. All other hyperparameters used in training the one-shot model are the same as in RSWS.

We run CoNAS in two different settings to find small and large size CNN cells. Specifically, we use the sparsity parameters $s = 10$, Fourier basis degree $d = 2$, and Lasso coefficient $\lambda = 1$ (We include experiments with varying lasso coefficients in Appendix A.6). As a result, we found the *normal cell* and *reduce cell* with one sparse recovery stage as shown in Appendix A.4 (the larger CNN cells were found with multiple sparse recovery stages). Repeating four stages of $(t = 4)$ of sparse recovery with restriction in definition (2.2) returns an architecture encoder $\boldsymbol{\alpha}^*$ with numerous operation edges in the cells (Please see Appendix A.9.1 for the found architecture). We support the effect of multiple stages sampling providing the empirical experiments on Appendix A.7. Now, we evaluate the model found by CoNAS as follows:

**Architecture Evaluation.** We re-train the final architecture with the learned cell and with the same hyperparameter configurations in DARTS to make the direct comparisons. We use NVIDIA TITAN X, GTX 1080, and Tesla V100 for final architecture training process. CoNAS cells from four sparse

recovery stages (t=4) cannot use the same minibatch size (i.e., 96) used in DARTs and RSWS, due to the hardware constraint; instead, we re-train the final model with minibatch size 56 with TITAN X. CoNAS architecture with one sparse recovery (t=1) outperforms DARTs and RSWS (stronger than vanilla random search) in test errors with smaller parameters, multiply-addition operations, and search time. In addition, CoNAS with four recovery stages (t=4) performs better than CoNAS (t=1) on both lower test error average and deviation; however, it requires larger parameters and multiply-add operations compared to DARTs, RSWS, and CoNAS (t=1). We also train CoNAS (t=1) with increasing the number of channels from 36 to 60 and training epochs from 600 to 1,000 together with a recent data augmentation technique called AutoAugment (Cubuk et al., 2019), which breaks through 2% test error barrier on CIFAR-10.

## 4.2 TRANSFER TO OTHER DATASETS

We test the cell found from CIFAR-10 to evaluate the transferability to different datasets: CIFAR-100, SVHN, and Fashion-MNIST in Table 2. As we can see, CoNAS achieves the competitive results with the smallest architecture size compared to the other algorithms.

## 4.3 PENN TREEBANK

**Architecture Search.** Similar to the setup in DARTS and RSWS, CoNAS explores the cell with the following operations: Tanh, ReLU, Sigmoid, and Identity. We augment the RNN cell with a variation of highway connections suggested by Pham et al Pham et al. (2018). Layers of depth $l$ in Recurrent Highway Network (Zilly et al., 2017) utilize a nonlinear transformation from its hidden state $h_l$ as follows:

$$h_l = (1 - c_l) \otimes \texttt{activation}(h_{l-1} \cdot W_{l,l-1}^{(h)}) + h_{l-1} \otimes c_l,$$

where "$\otimes$" denotes the element-wise multiplication. Since we use an expanded search space, we allow multiple operations; in such cases, we replace $\texttt{activation}(h_{l-1} \cdot W_{l,l-1}^{(h)})$ with its sum-pooled version, $\frac{1}{n} \sum_i^n \texttt{activation}^{\texttt{i}}(h_{l-1} \cdot W_{l,l-1}^{(h)})$.

Pre-training the above RNN using weight-sharing can create unstable results since the sub-graphs could have some internal nodes with no connections, leading to exploding gradients. One way to mitigate this issue is to increase the $p$-parameter in the Bernoulli sampling to enforce connectivity; however, this can significantly slow down the computations. Hence, we add an additional heuristic of randomly activating an edge to connect the intermediate node if the node does not have any input edge according to its architecture encoder $\alpha$.

After obtaining the one-shot model, we randomly sample the measurements of the sub-graph without the above heuristic used in the training stage. Running CoNAS with two stages of sparse recovery with $s = 10$ and $s = 5$ finds enough number of edges for the RNN cell. If the final resulting cells has intermediate nodes with a disconnected input, we added ReLU operations from the previous intermediate node. The visualization of the RNN cell found by CoNAS is shown in Appendix A.9.2.

Table 2: *Image Classification Test Error of CoNAS on Multiple Datasets. We compare the performance of CoNAS on different datasets with existing NAS results. The experiment details for CoNAS is described in Appendix A.5.*

| Architecture | CIFAR100 (%) | SVHN (%) | F-MNIST (%) | Params (M) | Search (GPU days) |
|---|---|---|---|---|---|
| SNAS[#] (Xie et al., 2018) | 16.5 | 1.98 | 3.73 | 2.8 | 1.5 |
| PNAS[#] (Liu et al., 2018a) | 15.9 | 1.83 | 3.72 | 3.2 | 150 |
| NASNet[#] (Zoph et al., 2018) | 15.8 | 1.96 | 3.71 | 3.3 | 1800 |
| DARTs[#] (Liu et al., 2018b) | 15.8 | 1.85 | 3.68 | 3.4 | 1 |
| AmoebaNet-A[#] (Real et al., 2019) | 15.9 | 1.93 | 3.8 | 3.2 | 3150 |
| ASAP[#] (Noy et al., 2019) | 15.6 | 1.81 | 3.73 | 2.5 | 0.2 |
| CoNAS (t=1) | 15.9 | 1.44 | 4.11 | 2.3 | 0.4 |

[#] This is the experimental result taken from Noy et al. (2019).

Table 3: ***Comparison of state-of-the-art NAS methods and hand-designed networks on PTB*** *(Lower perplexity is better). The results are grouped in following orders: manually designed networks, published NAS algorithms, and our experimental results. The average test error of our experiment used five random seeds. Table entries with "-" indicates either the field is not applicable or unknown.*

| Architecture | Test Perplexity | | Params (M) | Search Cost GPU days |
|---|---|---|---|---|
| | Valid | Test | | |
| Variational RHN (Zilly et al., 2017) | 67.9 | 65.4 | 23 | - |
| LSTM + DropConnect (Merity et al., 2018) | 60.0 | 57.3 | 24 | - |
| LSTM + Mos (Yang et al., 2018) | 56.5 | 54.4 | 22 | - |
| NAS (Zoph & Le, 2017) | - | 64.0 | 25 | 1e4 |
| ENAS[†] (Pham et al., 2018) | - | 56.3 | 24 | 0.5 |
| Random search[†] (Liu et al., 2018b) | 61.8 | 59.4 | 23 | 2 |
| DARTs (1st order)[†] (Liu et al., 2018b) | 60.2 | 57.6 | 23 | 0.5 |
| DARTs (2nd order)[†] (Liu et al., 2018b) | 58.1 | 55.7 | 23 | 1 |
| ASHA[*] (Li & Talwalkar, 2019) | 58.6 | 56.4 | 23 | - |
| RSWS[*] (Li & Talwalkar, 2019) | 57.8 | 55.5 | 23 | 0.25 |
| DARTs (2nd order)[+] | 60.7 | 58.0 | 23 | 1 |
| RSWS[#, +] | 60.6 | 57.9 | 23 | 0.25 |
| CoNAS[+] | 59.1 | 56.8 | 23 | 0.25 |

[#] We used the RSWS code with adjusting the search time equivalent to CoNAS.
[†] Used NVIDIA GTX 1080Ti GPU for training/searching.
[*] Used Tesla P100 GPU for training/searching.
[+] Used NVIDIA Titan X GPU for training/searching.

**Architecture Evaluation.** The results for recurrent architectures are presented in Table 3. We trained the final RNN model with the learned cell and the same hyperparameters in DARTs and RSWS, except with minibatch size equals 128 (due to hardware constraints). We also included the experimental results with RSWS methods allocating the equivalent search time with our methods to make a fair comparison with CoNAS. Since the published NAS literature in Table 3 uses different GPU hardware (e.g. DARTs and ENAS: NVIDIA GTX 1080Ti, RSWS: Tesla P100), a one-to-one comparison of the search cost value listed in Table 3 is not applicable. Our experiments show that CoNAS finds better performing architectures when compared with DARTs and RSWS; however, note that these differ slightly from the published experimental results for DARTs and RSWS, which we could not reproduce.

## 4.4 DISCUSSION

Noticeably, CoNAS achieves improved results on CIFAR-10 in both test error and search cost when compared to the previous state-of-the-art algorithms: DARTs, RSWS, and ENAS. In addition, not only CoNAS finds the cell with smallest parameter size and multiply-add operations than the other NAS approaches, but also it obtains a better test error with $2.57\%$. Many previous NAS papers have focused on the search strategy, while adopted the same search space to (Zoph et al., 2018) and (Liu et al., 2018b). Our experimental results highlight the importance of both seeking new performance strategies and the search space.

Finally, on PTB, our experiments show that CoNAS finds a better RNN architecture than RSWS, DARTs using an equivalent or less search cost. However, the reported test perplexity of DARTs and RSWS outperforms both valid and test perplexity of CoNAS.

## 5 PRIOR ART

We conclude by briefly reviewing the NAS literature and highlighting connections with CoNAS.

**Neural Architecture Search.** Early NAS approaches used RL-based controllers (Zoph et al., 2018), evolutionary algorithms (Real et al., 2019), or sequential model-based optimization (SMBO) (Liu et al., 2018a), and showed competitive performance with manually-designed architectures such as

deep ResNets (He et al., 2016) and DenseNets (Huang et al., 2017). However, these approaches required substantial computational resources, running into thousands of GPU-days. Subsequent NAS works have focused on boosting search speeds by proposing novel search strategies, such as differentiable search technique via gradient-based optimization (Cai et al., 2019; Liu et al., 2018b; Noy et al., 2019; Luo et al., 2018; Xie et al., 2018) and random search via sampling sub-networks from a one-shot supernetwork (Bender et al., 2018; Li & Talwalkar, 2019). Other recent NAS approaches include RL approaches via weight-sharing to boost speeds compare to vanilla RL (Pham et al., 2018), network transformations (Cai et al., 2018; Elsken et al., 2019; Jaderberg et al., 2017; Jin et al., 2018; Liu et al., 2017; Hu et al., 2019), and random exploration (Li et al., 2018; Li & Talwalkar, 2019; Sciuto et al., 2019; Xie et al., 2019). To the best of our knowledge, no NAS method yet reported has explored compressive sensing techniques.

**Differentiable Neural Architecture Search (DARTs).** Our CoNAS approach can be viewed as a refinement to DARTs ((Liu et al., 2018b)) which performs bilevel optimization by relaxing the (discrete) architecture search space to a differentiable search space via softmax operations. The choice of alternative optimization on differentiable multi-objective formulation substantially speeds up the search by orders of magnitude while achieving competitive performance compared to previous works (Zoph & Le, 2017; Zoph et al., 2018; Real et al., 2019; Liu et al., 2018a).

**One-Shot Neural Architecture Search.** Bender et al. (2018) provide an extensive experimental analysis on one-shot architecture search based on weight-sharing. Bender et al. (2018) statistically showed the correlation between the one-shot model (supergraph) and stand-alone model (subgraph) through the experiments. Li & Talwalkar (2019) proposes simplified training procedures without stabilizing techniques (e.g., path dropout schedule on a direct acyclic graph (DAG) and ghost batch normalization) from Bender et al. (2018). As the final performance of the discovered architecture heavily relies on hyperparameter settings, Li & Talwalkar (2019) exactly accords hyperparameters and data augmentation techniques to DARTs for their experiments. This combination of random search via one-shot models with weight-sharing provides the best competitive baseline results reported in the NAS literature. Our CoNAS approach improves upon these reported results.

**Learning Sub-Networks.** Stobbe & Krause (2012) propose learning sparse sub-networks from a small number of random cuts; they also leverage ideas from compressive sensing and provide theoretical upper bounds for successful recover. Our CoNAS approach is directly inspired from their seminal work. However, we emphasize essential differences: while Stobbe & Krause (2012) emphasize *linear* measurements, CoNAS takes a different perspective by focusing on measurements that map sub-networks to performance, which are fundamentally *nonlinear*. Moreover, our theoretical bounds use better Fourier-RIP bounds, and lead to improved results in terms of measurement complexity.

**Hyperparameter optimization.** Building upon the approach of Stobbe & Krause (2012), Hazan et al. (2017) develop a spectral approach called *Harmonica* for hyperparameter optimization (HPO) by encoding hyperparameters as binary strings. CoNAS also follows the same path, albeit for NAS. While NAS and HPO are sister meta-learning problems, we emphasize that our focus is exclusively on NAS, while Hazan et al. (2017) exclusively focus on HPO.

Moreover, the techniques of Hazan et al. (2017) cannot be directly applied to the NAS problem. We need to define our search space, encode our search problem in terms of Boolean variables, and propose how to gather measurements. All these are new to our paper: in particular, CoNAS proposes gathering measurements within tractable sampling time as described in Appendix A.1, while Harmonica naively gathers the approximated measurements by training the model for each randomly sampled hyperparameter choice. In Appendix A.1, we show that describes the sampling times with the naive sampling method for 1,000 measurements takes approximately 174 GPU days, whereas CoNAS only take 0.02 GPU days to gather 1,000 measurements. Finally, Harmonica requires invocation of a baseline hyperparameter optimization method (such as random search, successive halving (Jamieson & Talwalkar, 2016), or Hyperband (Li et al., 2017)), which CoNAS does not require.

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

## A  Appendix

### A.1  Supplementary Background of One-Shot Neural Architecture Search

In this section, we explain the one-shot neural architecture search in more details regarding of three directions introduced by Elsken et al. (2018).

**Search Space.**  First, we start with the search space, defining the principles of constructing neural architecture. As discussed before, simplifying the search space using human prior knowledge can help search algorithms to find an optimal candidate faster. However, it may limit the algorithm to find a novel architecture beyond our knowledge due to human bias. One-shot neural architecture search methodology is not restrict to the specific search space (e.g. RSWS (Li & Talwalkar, 2019) uses the equivalent search space to DARTs (Liu et al., 2018b)).

**Search Strategy**  Next, we have a search strategy which is considered as a methodology to find the best neural architecture. In the last two years, different methodologies for search space such as reinforcement learning, evolutionary algorithms, SMBO, Bayesian optimization, bilevel optimization, and randomness are introduced in NAS literature. For instance, one-shot architecture uses random search as a search strategy.

**Performance Estimation Strategy.**  One-shot NAS aims to reduce the computational cost for the searching with the surrogate model to estimate the sub-networks performance. The conventional performance estimating methods require the massive computational resources as each model need to be trained and evaluated separately. In particular, assume that our goal is to construct a CNN by using a proxy architecture through randomly selected cell samples by concatenating eight layers with 16 initial channels (This is the same proxy model setup in DARTs). Then, we measure the validation loss from 50 epochs trained model where each epoch takes five minutes of training. As a result, the total sampling time to collect 1,000 randomly sampled of the architecture performance (which is the same number of measurements as our experiment using CoNAS with $t = 1$) equals to $5 \cdot 50 \cdot 1000 \cdot \frac{1}{60} \cdot \frac{1}{24} \approx 174$ days.

Instead, one-shot NAS trains the super-network with weight-sharing by randomly sample a sub-network and only update the weights corresponding to the sampled network for each minibatch. Bender et al. (2018) demonstrates the correlation between the performance of one-shot model and the stand-alone architectures. Moreover, Li & Talwalkar (2019) provides the experimental results that the random search with weight-sharing is a competitive baseline even outperforming ENAS (Pham et al., 2018).

### A.2  Prior Works on Recovery Conditions on Compressive Sensing

There has been significant research during the last decade in proving upper bounds on the number of rows of bounded orthonormal dictionaries (matrix $\mathbf{A}$) for which $\mathbf{A}$ is guaranteed to satisfy the restricted isometry property with high probability. One of the first BOS results was established by Candes & Tao (2006), where the authors proved an upper bound scales as $\mathcal{O}(sd^6 \log^6 n)$ for a subsampled Fourier matrix. While this result is seminal, it is only optimal up to some *polylog* factors. In fact, the authors in chapter 12 of Foucart & Rauhut (2017) have shown a necessary condition (lower bound) on the number of rows of BOS which scales as $\mathcal{O}(sd \log n)$. In an attempt to achieve to this lower bound, the result in Candes & Tao (2006) was further improved by Rudelson & Vershynin (2008) to $\mathcal{O}(sd \log^2 s \log(sd \log n) \log n)$. Motivated by this result, Cheraghchi et al. (2013) has even reduced the gap further by proving an upper bound on the number of rows as $\mathcal{O}(sd \log^3 s \log n)$. The best known available upper bound on the number of rows appears to be $\mathcal{O}(sd^2 \log s \log^2 n)$; however with worse dependency on the constant $\delta$, i.e., $\delta^{-4}$ (please see Bourgain (2014)). To the best of our knowledge, the best known result with mild dependency on $\delta$ (i.e., $\delta^{-2}$) is due to Haviv & Regev (2017), and is given by $\mathcal{O}(sd \log^2 s \log n)$. We have used this result for proving Theorem 3.2.

### A.3  CoNAS Algorithm

In this section, we provide the pseudocode of CoNAS explained in the Section 3 from the main paper.

---

**Algorithm 1** CoNAS

---

1: **Inputs:** Number of one-shot measurements $m$, stage $t$, sparsity $s$, lasso parameter, $\lambda$, Bernoulli $p$

---

**Stage 1 – Training the One-Shot Model**

---

2: **procedure** MODEL TRAINING
3:     **while** not converged **do**
4:         Randomly sample a sub-architecture encoded binary vector $\boldsymbol{\alpha}$ according to Bernoulli($p$)
5:         Update weights $w_{\boldsymbol{\alpha}}$ by descending $\nabla_{w_{\boldsymbol{\alpha}}} \mathcal{L}_{train}(w_{\boldsymbol{\alpha}})$
6:     **end while**
7: **end procedure**

---

**Stage 2 – Search Strategy**

---

8: **procedure** ONE-SHOT MODEL APPROXIMATION VIA COMPRESSIVE SENSING
9:     **for** $k \in \{1, \ldots, t\}$ **do**
10:        Collect $\mathbf{y} = (f(\boldsymbol{\alpha}_1), f(\boldsymbol{\alpha}_2), \ldots, f(\boldsymbol{\alpha}_m))^\top$.
11:        Solve

$$\mathbf{x}^* = \arg\min_{\mathbf{x}} \|\mathbf{y} - \mathbf{A}\mathbf{x}\|_2^2 + \lambda \|\mathbf{x}\|_1$$

12:        Let $x_1^*, x_2^*, \ldots, x_s^*$ be the $s$ absolutely largest coefficients of $\mathbf{x}^*$. Construct

$$g(\boldsymbol{\alpha}) = \sum_{i=1}^{s} x_i^* \chi_i(\boldsymbol{\alpha})$$

13:        Compute minimizer $\mathbf{z} = \arg\min_{\boldsymbol{\alpha}} g(\boldsymbol{\alpha})$ and let $\overline{J}$ the set of indices of $z$.
14:        $f = f_{J|\mathbf{z}}$
15:     **end for**
16:     Construct the cell by activating the edge where $z_i = 1$ where $i \in [n]$.
17: **end procedure**

---

### A.4 ARCHITECTURE FOUND FROM CoNAS FOR IMAGE CLASSIFICATION TASK

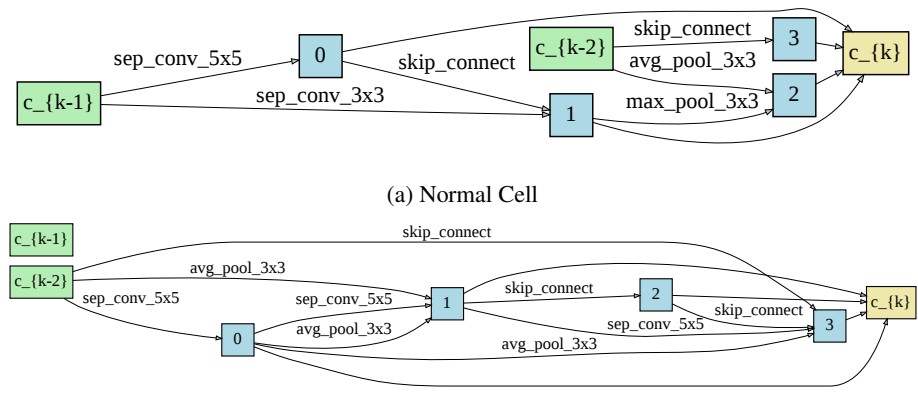

(a) Normal Cell

(b) Reduce Cell

Figure 3: *Convolution Cell found from CoNAS. The reduce cell found from CoNAS have a missing connection between $c_{k-1}$ and intermediate nodes which is a valid architecture in our search space.*

### A.5 TRAINING DETAILS ON OTHER DATASETS

**CIFAR-100** This dataset is extended version of CIFAR-10 with 100 classes containing 600 images each. Similar to CIFAR-10, CIFAR100 consists of 60,000 color images which splits into 50,000 training images and 10,000 test images. Following the existing works (Liu et al., 2018b), we train the architecture with 20 stacked cells equivalent to CIFAR-10 setting. We train the architecture for

600 epochs with cosine annealing learning rate where the initial value is 0.025. We use a batch size 96, SGD optimizer with nestrov-momentum of 0.9, and auxiliary tower with weights 0.4. For the regularization technique, we include path dropout with probability 0.2, cutout regularizer with length 16, and AutoAugment (Cubuk et al., 2019) for CIFAR-100. Except AutoAugment, the training setup is identical to DARTs for CIFAR-10.

**Street View House Numbers (SVHN)**  SVHN is a digit recognition dataset of house numbers obtained from Google Street View images. SVHN consists of 73,257 train digit images, 26,032 test digit images, and additional 531,131 images. We used both train and extra (total 604,388) images for the training the architecture. Due to the large dataset, we train the architecture for 160 epochs (equivalent to ) and other hyperparameter setup is equivalent to CIFAR-100.

**Fashion-MNIST**  Fashion-MNIST concists of grayscale 60,000 train image set and 10,000 test image set with the size $28 \times 28$ associated with 10 classes of labels. Training hyperparameter setup of the final architecture is equivalent to CIFAR-10 without AutoAugment (Cubuk et al., 2019).

### A.6   STABILITY ON LASSO PARAMETERS

We check our algorithm's stability on lasso parameter by observing the solution given exact same measurements. Denote $\alpha^*_{\lambda=l}$ as the architecture encoded output from CoNAS given $\lambda = l$. We compare the hamming distance and the test error between $\alpha^*_{\lambda=1}$ and other $\lambda$ values ($\lambda = 0.5, 2, 5, 10$). The average support of the solution from one sparse recovery is 15 out of the 140 length. The average hamming distance between two randomly generated binary strings with $\text{supp}(\alpha^*) = 15$ from $100,000$ samples was $27.58 \pm 1.82$. Our experiment shows a stable performance under various lasso parameters with small hamming distances regards to various $\lambda$. Also we measure the average test error with 150 training epochs on different $\lambda$ values as shown in Table 4. For the baseline comparison, we compare CoNAS solutions with the randomly chosen architecture with 15 operations.

Table 4: Lasso Parameter Stability Experiment.

| Criteria | $\lambda = 0.5$ | $\lambda = 2.0$ | $\lambda = 5.0$ | $\lambda = 10.0$ | Random |
|---|---|---|---|---|---|
| Hamming Distance | 0 | 0 | 8 | 12 | 29 |
| Test Error (%) | $3.74 \pm 0.07$ | $3.74 \pm 0.07$ | $3.51 \pm 0.06$ | $3.62 \pm 0.04$ | $4.43 \pm 0.08$ |
| Param (M) | 2.3 | 2.3 | 2.6 | 2.6 | 2.7 |
| Multiply-Add (M) | 386 | 386 | 455 | 449 | 444 |

## A.7 EFFECT OF MULTIPLE STAGES SAMPLING

We illustrate an experiment which shows that the CoNAS with multiple stages can successively discover important edges, and eventually find a architecture with smaller loss/perplexity. Suppose we sample 1,000 measurements and computed the mean and the standard deviation of the validation loss from CIFAR-10 (perplexity from PTB) for each stage. As we can see in Figure 4, the larger number of stages, the smaller loss/perplexity. While the multiple stage sampling finds more operation edges in the cell (equivalent to finding larger sub-graph), it may increase the architecture size (in terms of parameters and multiply-addition operation) as shown in Table 1.

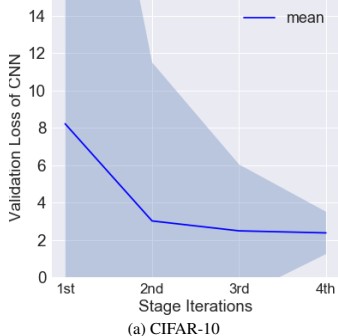 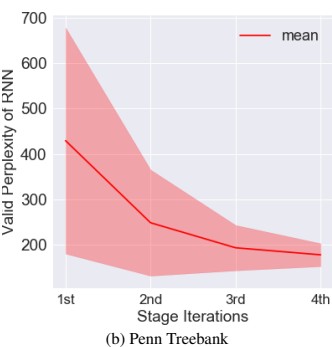

(a) CIFAR-10        (b) Penn Treebank

Figure 4: *The one-shot architecture validation loss/perplexity vs stage iterations of (a) CIFAR-10 and (b) PTB. The line plot and shaded region correspond to the average and standard deviation of measurements respectively.*

## A.8 EFFECT OF INCREASING OPERATION EDGES IN THE CELL.

We provide more experiments on training results of randomly wired cells which have a similar number of edges to the cell found by CoNAS (t=4). We randomly sampled the architecture code $\alpha$ with $Binomial(140, 1/4)$ for each digit being 1 that the expected number of total edges in *normal cell* and *reduce cell* is 35. We randomly selected three architectures as shown in Appendix A.9.5 and trained with the same settings to the section 4.1. Table 5 shows that choosing a larger number of randomly chosen edges is not sufficient to improve the model's performance.

Table 5: **Randomly Wired Model Performance on CIFAR-10**. (Lower test error is better) Trained with auxiliary towers and cutout augmentation for 600 epochs (equivalent training setup to CIFAR-10 from DARTs.

| Method | Number of Edges | Size (M) | Test Error |
|---|---|---|---|
| Randomly Wired Model (1) | 31 | 5.2 | 3.45% |
| Randomly Wired Model (2) | 35 | 4.5 | 2.89% |
| Randomly Wired Model (3) | 32 | 3.1 | 3.52% |

### A.9    ARCHITECTURE FOUND FROM OUR EXPERIMENT

#### A.9.1    COMPRESSIVE SENSING-BASED NEURAL ARCHITECTURE SEARCH (CONAS) FOR CNN WITH FOUR STAGES OF SPARSE RECOVERY

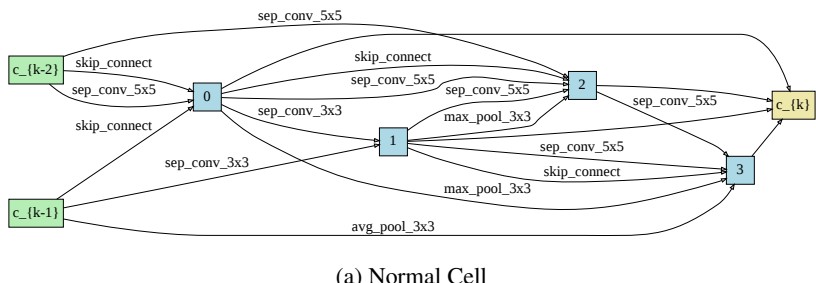

(a) Normal Cell

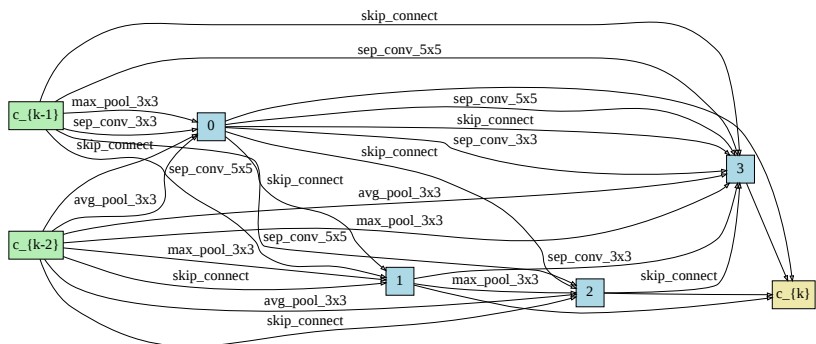

(b) Reduce Cell

Figure 5: Convolution Cell found from CoNAS (t=4)

#### A.9.2    COMPRESSIVE SENSING-BASED NEURAL ARCHITECTURE SEARCH (CONAS) FOR RNN

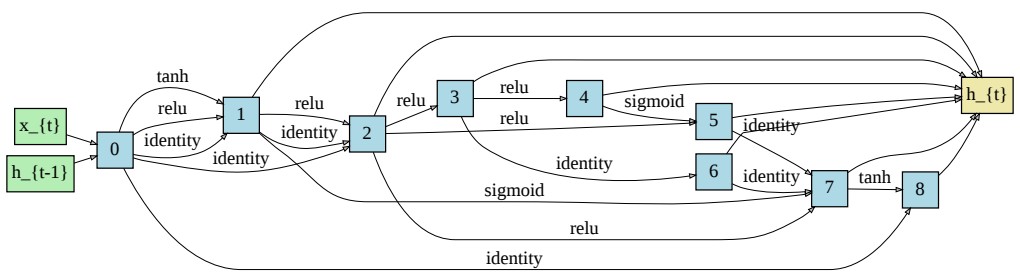

Figure 6: Recurrent Cell found from CoNAS

### A.9.3 DIFFERENTIABLE NEURAL ARCHITECTURE SEARCH (DARTS) FOR CNN

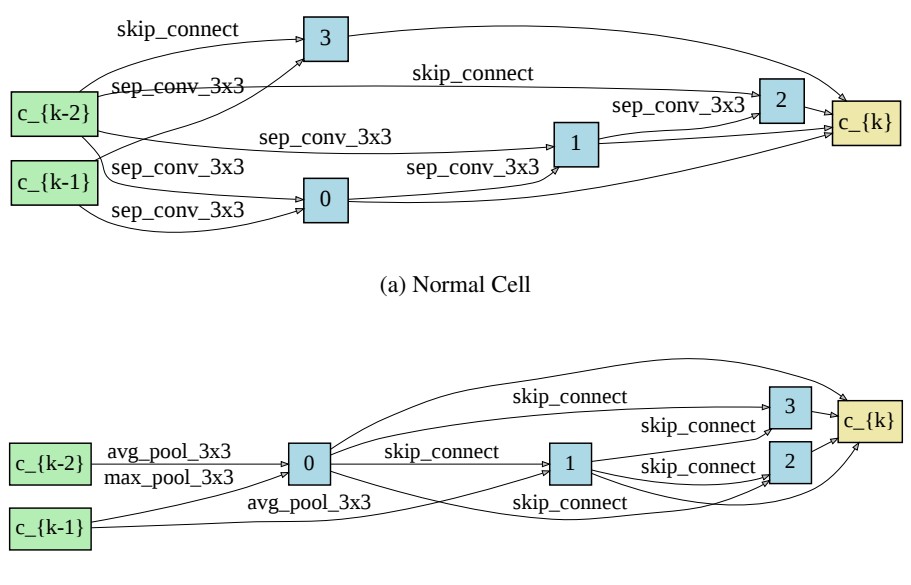

(a) Normal Cell

(b) Reduce Cell

Figure 7: Convolutional Cell found from DARTs with the original setting in Liu et al. (2018b).

### A.9.4 CoNAS SOLUTIONS WITH DIFFERENT LASSO PARAMETERS

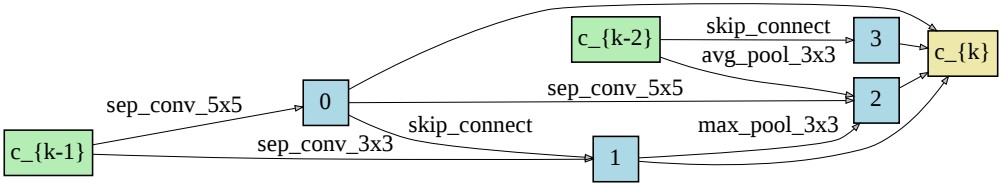

(a) Normal Cell with $\lambda = 5$

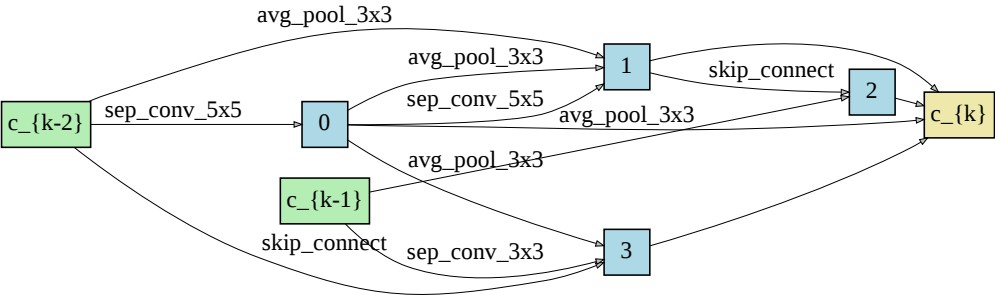

(b) Reduce Cell with $\lambda = 5$

Figure 8: CoNAS cell with $\lambda = 5$ corresponding to Table 4.

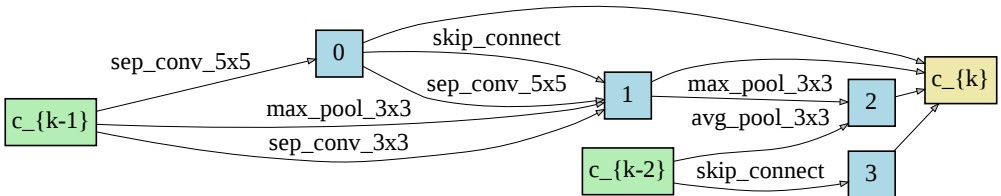

(a) Normal Cell with $\lambda = 10$

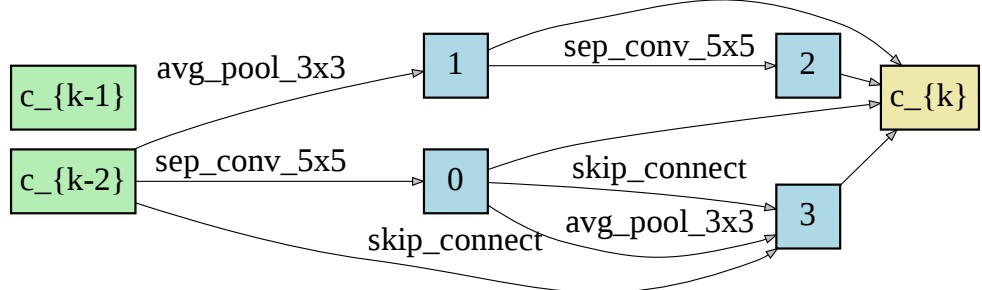

(b) Reduce Cell with $\lambda = 10$

Figure 9: CoNAS cell with $\lambda = 10$ corresponding to Table 4.

### A.9.5 RANDOMLY WIRED CNN ARCHITECTURES

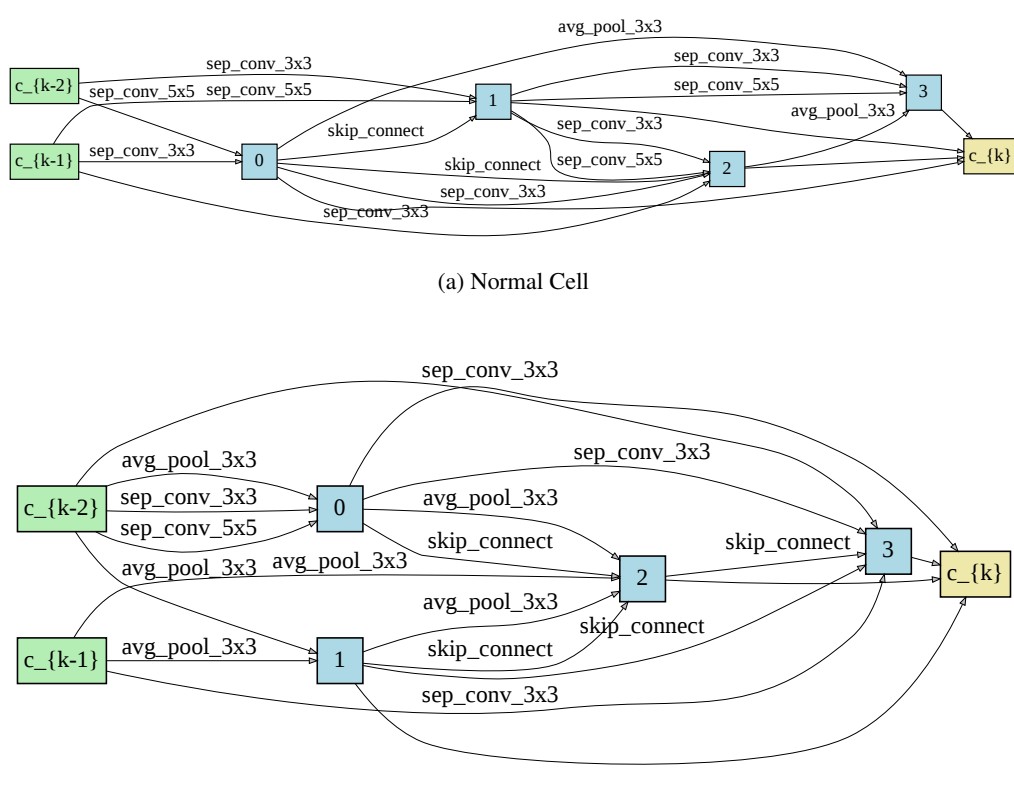

(a) Normal Cell

(b) Reduce Cell

Figure 10: Randomly wired cell corresponding to the first row result to Table 5.

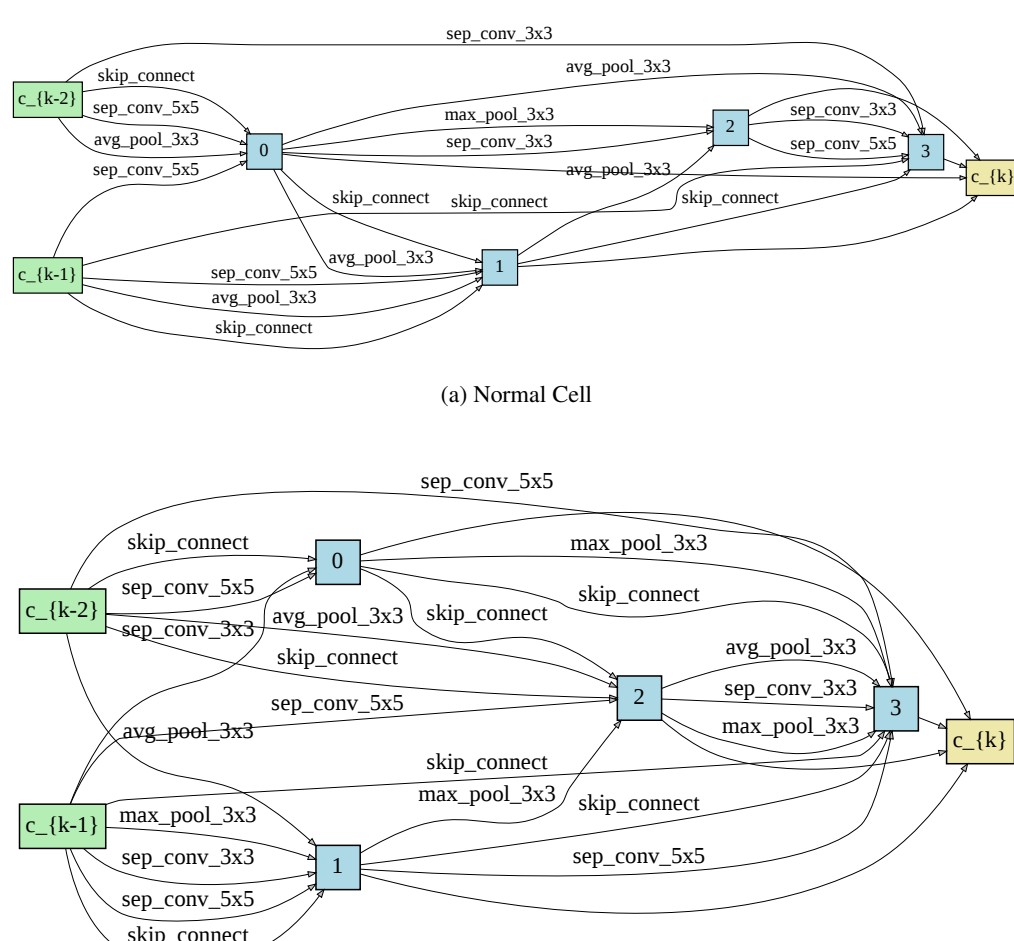

(a) Normal Cell

(b) Reduce Cell

Figure 11: Randomly wired cell corresponding to the second row result to Table 5.

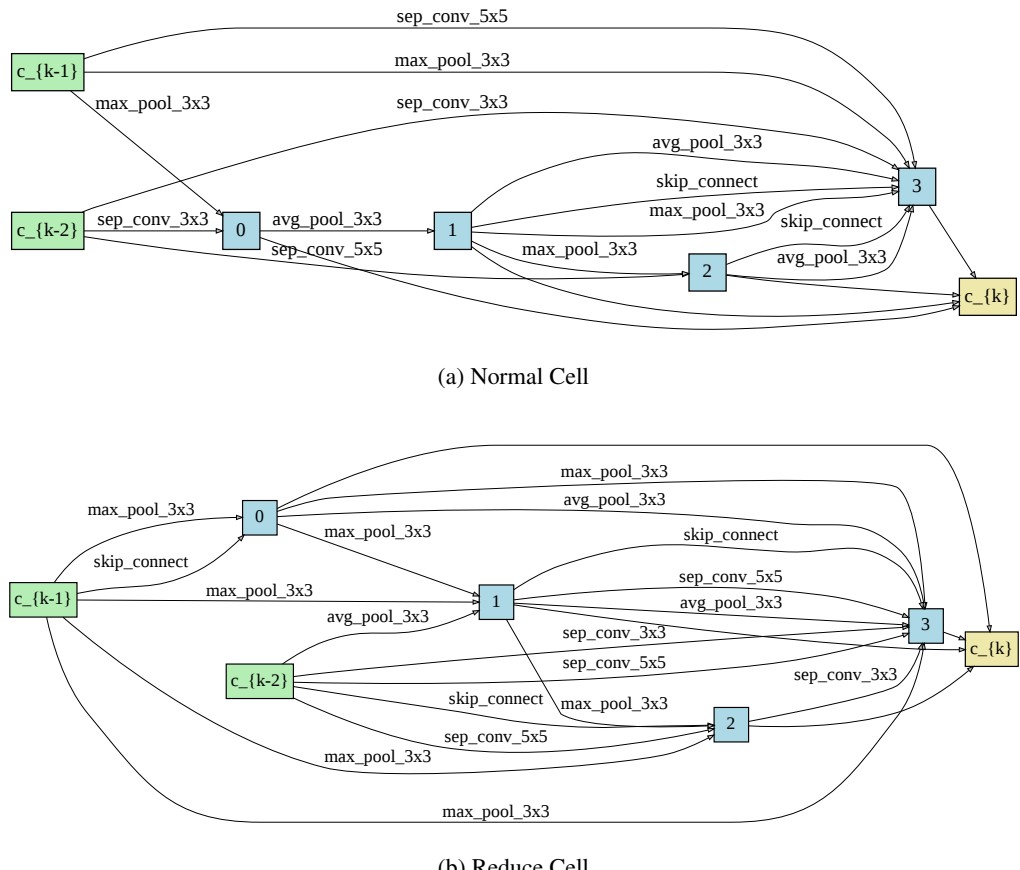

(a) Normal Cell

(b) Reduce Cell

Figure 12: Randomly wired cell corresponding to the third row result to Table 5.

A.9.6 RANDOM SEARCH WITH WEIGHT-SHARING (RSWS) FOR RNN

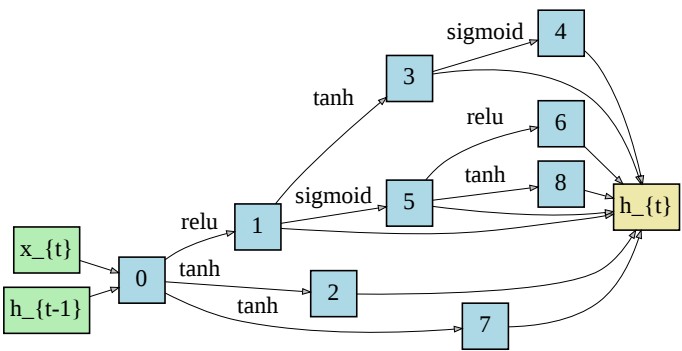

Figure 13: Recurrent cell found from RSWS allocating equal amount of search time to CoNAS

