# OpenReview forum: "One-Shot Neural Architecture Search via Compressive Sensing"
_ICLR.cc/2020/Conference — Reject_

### Official Review · AnonReviewer1 · 2019-10-14
**Official Blind Review #1050**

**Rating:** 1

**Review:**

In this paper, the authors study Neural Architecture Search, which aims to automate design of neural network models. Their approach consists in using a two stage algorithm:
-	A first Neural Network $f$ is trained for predicting the performances of sub-architectures. Then binary sub-graphs coding the sub-architectures are uniformly sampled (Bernouilli(0.5)), and their performances y are evaluated thanks to the first Neural Network.
-	The graph sampling matrix A, which is indexed by the m sampled architectures and the Fourier basis of size $O(n^d)$, is built. Then the optimization problem $x^*= \arg\min_x ||y – Ax||$ is solved using Lasso. The largest Fourier coefficients are chosen to build an estimate g of f. Finally, computing minimum of g, the architecture is generated.

Since $m << n^d$, the optimization problem is ill-posed. Theorem 3.2 shows that if A satisfies the restricted isometry of order s, then the sparse coefficients x can be recovered.
The algorithm is evaluated and compared to the state-of-the-art on various image classification tasks and on RNN.


Major concern:

1/ I did not find the proof of Theorem 3.2 in the main paper and in the appendix, so I do not buy it.

2/ The authors claim that their algorithm performs better than the state-of-the-art, but according to tables 1,2,3, I did not find significant differences of performances in term of test errors.

3/ The key idea of the algorithm is not well explained. A one-shot NAS f is pre-trained. f is assumed to be well-trained. Then it is approximated with a Fourier-sparse Boolean function. Why using an approximation if f is perfect? Do you expect to reduce the needed number of sampled architectures?


Minor concerns:

Equation 3.1 is not clear. $X_S (\alpha_l)$ does not depend on k. So all rows seem to be the same.



**Experience Assessment:**

I do not know much about this area.

**Review Assessment: Checking Correctness Of Derivations And Theory:**

I carefully checked the derivations and theory.

**Review Assessment: Checking Correctness Of Experiments:**

I assessed the sensibility of the experiments.

**Review Assessment: Thoroughness In Paper Reading:**

I read the paper at least twice and used my best judgement in assessing the paper.

---

> ### Author Response · Authors · 2019-11-11
> **Author Response to Official Blind Review #1**
>
> Thank you for your review. We've added corrections and clarifications regards to your concern on our paper.
>
> Q1: We thank the reviewer for raising this point. In the submitted version, we had provided some discussion regarding the upper bound of the number of rows for the graph-sampling matrix A to satisfy RIP. We have clarified the proof right after Theorem 3.2 in the paper.
>
>
> Q2: We agree that the improvement of test error is not significantly different than the current state-of-the-art. However, we respectfully push back against the criticisms of lack of improvement. The criteria for evaluating algorithm performance in NAS involves not only the test error, but also the search time, model size, and number of FLOPs.
>
> The seminal work of NAS starts from Zoph et al. [1] suggests the method of automatically designing the competitive architectures (test error: 2.65%, model size: 3.3M on CIFAR-10), but required substantial computational resources (2000 GPU days). Numerous later works have been introduced to reduce the search time, but the test error performance has remained comparable to [1]. These are summarized in this table:
>
> ---------------------------------------------------------------
> |Method     |Test Error(%)   |Search Time (Days/GPU)|Model Size (M)|
> |NASNET[1]|2.65%               |2000                                  |3.3                   |
> |ENAS  [2]   |2.89%               |0.45                                   |4.6                   |
> |DARTS [3]  |2.76 \pm 0.09  |4                                        |3.3                   |
> |SNAS  [4]   |2.85 \pm 0.02  |1.5                                     |3.3                   |
> |RSWS  [5]  |2.85 \pm 0.08  |2.7                                     |3.7                   |
> |GHN   [6]   |2.84 \pm 0.07  |0.84                                   |5.7                  |
> ---------------------------------------------------------------
>
> We would like to highlight that a perfectly fair comparison between the performance of NAS algorithms are currently very challenging since various hyperparameters are involved in the training protocol for the final evaluation: cutout [7], auto-augment [8]), training epochs, learning rate decay methods ([9]). We refer to Table 1, which shows CoNAS breaks the 2% barrier of test error on CIFAR-10 with auto-augment.
>
> Our focus, therefore, has been to show the impact of a better search strategy (Fourier sparse recovery) while keeping other factors as similar as possible. The hyperparameter setup (training protocol for the final evaluation) of CoNAS is exactly the same as that of DARTs, but we demonstrate that CoNAS achieves a (small) improvement in test error but given smaller model size, lesser number of FLOPs, and  significantly reduced search time.
>
>
> Q3: The main idea of our algorithm is to represent the one-shot model (f) with its Boolean expansion; note that this trick works for any $f$ that is a black-box function. We note that the vanilla search strategy with one-shot architecture randomly samples the architectures and pick the best one with the smallest loss (random search), while the objective of our method is to express $f$ in terms of the polynomial basis functions ($g$).
>
> We would like to point out that CoNAS achieved better performance than random search with weight-sharing [5] while the search time of CoNAS is much shorter than [5]: 0.4 Day/GPU for our method and 2.7 Day/GPU for [5] (the longer search time allows to sample more architecture.
>
>
> Minor concern: We gratefully thank the reviewer for pointing out the error in Equation 3.1. In the graph-sampling matrix A, the row corresponds to the measurement while the columns correspond to the Fourier basis. We have fixed the expression accordingly.
>
> Again, we thank your time and effort for reviewing our paper.
>
>
> References
> [1] Barret Zoph, Vijay Vasudevan, Jonathon Shlens, Quoc V. Le. Learning Transferable Architectures for Scalable Image Recognition. CVPR 2018.
> [2] Hieu Pham, Melody Y. Guan, Barret Zoph, Quoc V. Le, Jeff Dean. Efficient Neural Architecture Search via Parameter Sharing. In ICML, 2018.
> [3] Hanxiao Liu, Karen Simonyan, and Yiming Yang. DARTS: Differentiable architecture search. In ICLR, 2019.
> [4] Sirui Xie, Hehui Zheng, Chunxiao Liu, Liang Lin. SNAS: Stochastic Neural Architecture Search. ICLR 2019.
> [5] LIAM LI, AMEET TALWALKAR. Random Search and Reproducibility for Neural Architecture Search
> [6] Chris Zhang, Mengye Ren, Raquel Urtasun, Graph HyperNetworks for Neural Architecture Search. ICLR 2019
> [7] Terrance DeVries, Graham W. Taylor. Improved Regularization of Convolutional Neural Networks with Cutout, 2017
> [8] Ekin D. Cubuk, Barret Zoph, Dandelion Mane, Vijay Vasudevan, Quoc V. Le. AutoAugment: Learning Augmentation Policies from Data, CVPR 2019
> [9] Andrew Hundt, Varun Jain, Gregory D. Hager. sharpDARTS: Faster and More Accurate Differentiable Architecture Search, 2019

---

### Official Review · AnonReviewer3 · 2019-10-22
**Official Blind Review #3**

**Rating:** 3

**Review:**

Contributions:
This paper tackles the problem of One-shot Neural architecture search by proposing a new method.
The method consists mainly of new search strategy of the optimal architecture that is inspired by the recovery of boolean functions from their sparse Fourier expansions. As such, this work is an application of recent progress in the field of compressive sensing to One-shot neural architecture search. Given the problem formalism, the authors have also provides guarantee for the optimality of their method, i.e the method can recover the optimal sub-network of any given  a sufficient number of performance measurements.

Clarity
Overall, the paper is well motivated and the technical content is good. That said the structure could be enormously  improved to ease the reading and the overall understanding. For example: better caption for Figure 2 explaining what is shown; presenting the pseudo-code directly in the method overview and spending the rest of the section explaining the method; showing the related work before the experiments; etc.

Novelty
The main novelty in my opinion is the application of compressive sensing methods to One-shot NAS. This approach is significantly different from other One-shot NAS method that I am aware of mainly regarding the search strategy employed to find the best architecture.
However, this work seems like an incremental improvement over Hazan et al 2018. To this regard, the only novelty that was the framing of One-shot NAS as a recovery of boolean functions from their sparse Fourier expansions is not new either.  That is said, I am open to be proven wrong on this point!

Results
The experiment section is not self-content, the readers is refered a couple of times to other papers to get details that are  critical to reproducibility and understanding.
Overall, the search strategy of CoNAS seems  parameter efficient, fast and competitive. Also, small ablation studies showing the effect of the different parameters of CoNAS were very informative and well-appreciated.
However,  the search space is different between CoNAS and the others methods for some experiments, making it difficult to decide if the search strategy of CoNAS is definitely competitive compared to other methods or not.

Points of improvement:
1 - Structure of the paper
2 - Clarify novelty compared to HARMONICA ( not the application domain please)
3 - demonstrate that with the same search space your method is competitive.
4 - Does m=1000 in your experiments satisfies theorem 3.2? What is the value of d in your experiments? Can you provide supporting experiments that answer those questions?

Preliminary decision:
For now, I will say *weak reject*



**Experience Assessment:**

I have read many papers in this area.

**Review Assessment: Checking Correctness Of Derivations And Theory:**

I assessed the sensibility of the derivations and theory.

**Review Assessment: Checking Correctness Of Experiments:**

I assessed the sensibility of the experiments.

**Review Assessment: Thoroughness In Paper Reading:**

I read the paper at least twice and used my best judgement in assessing the paper.

---

> ### Author Response · Authors · 2019-11-11
> **Author Response to Official Blind Review #3**
>
> Thank you for your review! We have added the clarification regard to your concerns on our paper in below.
>
> Q1: We gratefully thank the reviewer for suggesting the structure of the paper. We have reorganized the paper accordingly to compress the main paper in 9 pages.
>
>
> Q2: We agree with the reviewer that Boolean function expansion using Fourier methods is a classical trick, used by multiple papers in the ML literature (e.g., Stobbe et al. [1], Kocaoglu et al. [2]) that predate the Harmonica work.
>
> We also agree that CoNAS follows the same path as Harmonica. As we explain in detail in Section 5, NAS and hyper-parameter optimization are both meta-learning problems, but the two bodies of literature in these two sub-fields of meta-learning have been largely disjoint.
>
> But beyond this, CoNAS also is different from Harmonica in two other important ways. Our measurements are gathered in a much more efficient manner (see Appendix A.1) leading to an improvement of nearly 4 orders of magnitude over the naive random sampling approach adopted by Harmonica, and our algorithm does not require invocation of a baseline HPO scheme like random search, successive halving, or Hyperband.
>
>
> Q3: We appreciate the reviewer for raising this point. As our paper mainly focuses on comparing our algorithm to DARTs with the fair comparison, we respectfully claim that the main factor of performance improvements comes from our search strategy.
>
> Our search space is a generalized version of proposed in DARTs [3] with larger capacity of possible architecture candidates. In other words, our search space is a superset of DARTs’ search space. Given an equivalent number of edges activated, the number of possible architectures from CoNAS is $\binom{140}{16} \approx 4.3 * 10^{20}$, while DARTs search space has approximately $10^{18}$ architecture candidates. The actual number of candidate architecture can in fact be larger than $4.3 * 10^{20}$ since the number of activated edges can vary depending on the solution of the sparse recovery step.
>
> The NAS survey paper [4] shows that reducing the size of the search space by incorporating human prior knowledge may accelerate the search time; however, this prevents finding novel architecture beyond the current human knowledge due to the introduction of human bias. We would like to point out that our search algorithm finds a winning architecture with smaller search time than [3] (0.4 days vs 4 days) while the search space is larger than that of [3].
>
>
> Q4: In our experiments, the values of different parameters are set to d=2, s=10, n =140. According to the bound provided in Theorem 3.2 for the number of rows of matrix A (i.e.,  m = O(s d log^2 s log n)), choosing m=100 will be consistent with these set of parameters. We have also explained this in the proof of Theorem 3.2.
>
>
> Again, we appreciate your time and effort looking into our paper.
>
> References
> [1] Peter Stobbe, Adreas Krause. Learning Fourier Sparse Set Functions, AISTAT 2012.
> [2] Murat Kocaoglu, Karthikeyan Shanmugam, Alexandros G. Dimakis, Adam Klivans. Sparse Polynomial Learning and Graph Sketching, NIPS 2014.
> [3] Hanxiao Liu, Karen Simonyan, and Yiming Yang. DARTS: Differentiable architecture search. In ICLR, 2019.
> [4] Thomas Elsken, Jan Hendrik Metzen, Frank Hutter. Neural Architecture Search: A Survey, 2019

---

### Official Review · AnonReviewer4 · 2019-11-05
**Official Blind Review #4**

**Rating:** 3

**Review:**

This paper proposes a new algorithm for one-shot neural architecture search (NAS) via compressive sensing. The authors propose a new search strategy, as well as a slightly different search space compared to DARTS [1], ProxylessNAS [2], etc. They use architecture samples from the one-shot model evaluated with the search parameters as a surrogate of the true objective in order to speed-up the search. Afterwards, these surrogate function evaluations are used to compute Fourier coefficients which are eventually used to optimize the vector of binary parameters encoding the architecture.

Overall, I think the proposed algorithm is interesting and of practical usefulness. However, in terms of novelty, this work seems more to be an application of Harmonica [6] to the NAS problem (with small modifications in order to make it applicable). In page 10 you state some of the differences of your method with Harmonica. I agree that the number of function evaluations you use (coming from the one-shot model) is larger and computationally less expensive to obtain, however this does not guarantee that these are a good surrogate of the true objective that NAS aims to minimize, i.e. the validation/test accuracy of final (stand-alone) architectures . The empirical evaluations of their algorithm seem to outperform/be competitive compared to other NAS methods on all benchmarks used in the paper, however only DARTS is evaluated on their search space and the other results are taken from the corresponding papers. The paper is well-written and -structured with the caveat of being more than the recommended 8 pages.

I will adjust my score depending on the authors responses concerning the following questions/issues:

1. The correlation between the architectures evaluated using the one-shot weights and retrained from scratch, seems to be of crucial importance in your method, since you directly use the one-shot weights to collect the measurements, similarly to Random Search with weight sharing [3], ENAS [4] or Bender et al. [5]. What is the correlation of these measurements with the stand-alone architectures trained from scratch using the final evaluation settings? How did you tune the p in the Bernoulli distribution during the one-shot weight updates. According to Bender et al. [5] the ScheduledDropPath probability is an important hyperparameter affecting the aforementioned correlation.

2. What is the main motivation for using 5 operations in the operation set and not 8 as in DARTS [1] for example? Does the main contribution in the competitive results come from the different search space or the search method?

3. Is there any reference or proof for the correctness of Theorem 3.2?

4. I think there are some parts that can be moved in the Supplementary, such as the pseudocode for the proposed algorithm or Figure 3, and some other parts that can be compressed, such as the Related Work section.

References
[1] Hanxiao Liu, Karen Simonyan, and Yiming Yang.  DARTS: Differentiable architecture search.  In ICLR, 2019.
[2] Han Cai, Ligeng Zhu, Song Han. ProxylessNAS: Direct Neural Architecture Search on Target Task and Hardware. In ICLR, 2019.
[3] LIAM LI, AMEET TALWALKAR. Random Search and Reproducibility for Neural Architecture Search.
[4] Hieu Pham, Melody Y. Guan, Barret Zoph, Quoc V. Le, Jeff Dean. Efficient Neural Architecture Search via Parameter Sharing. In ICML, 2018
[5] Gabriel Bender, Pieter-Jan Kindermans, Barret Zoph, Vijay Vasudevan, Quoc Le. Understanding and Simplifying One-Shot Architecture Search. In ICML, 2018
[6] Elad Hazan, Adam Klivans, Yang Yuan. Hyperparameter Optimization: A Spectral Approach


**Experience Assessment:**

I have published one or two papers in this area.

**Review Assessment: Checking Correctness Of Derivations And Theory:**

I assessed the sensibility of the derivations and theory.

**Review Assessment: Checking Correctness Of Experiments:**

I carefully checked the experiments.

**Review Assessment: Thoroughness In Paper Reading:**

I read the paper at least twice and used my best judgement in assessing the paper.

---

> ### Author Response · Authors · 2019-11-11
> **Author Response to Official Blind Review #4**
>
> We thank the reviewer for the insightful comments and appreciate the time and effort the Reviewer have given towards their reviews. Below we provide our responses to your concerns.
>
> Q1:
> We agree that the measurements from the stand-alone architecture and from the one-shot model must be strongly correlated if the performance estimation strategy is to succeed. However, we would like to point out that the one-shot model has been addressed multiple times in the NAS literature already. Bender et al. have studied this before: see Figure 5 in [1] for concrete support of this fact, showing strong monotonic correlation between the performance of a one-shot model trained with weight-sharing and a stand-alone model.
>
> Moreover, Random Search with weight sharing [2] simplifies the training procedure of the one-shot model from [1] omitting the complex stabilization techniques adopted in [1] such as ScheduledDropPath and ghost batch normalization, while achieving competitive results to existing methods. Our methodology follows the training algorithm from [2], and hence does not require the ScheduledDropPath probability hyperparameter. (See also Section 3 from [2].)  We specifically chose $p=0.5$ to train the one-shot model such that the one-shot model was optimized for the sub-network sampled from $\alpha ~ \{-1, 1\}^n$ under uniformly chosen random tuples.
>
>
> Q2: We thank the reviewer for raising this question. In the revision, we will clarify that the CoNAS performance improvement may come from both the search space and the search method, but we explain below that ultimately the final metric should be running time for a given accuracy.
>
> We agree that apples-to-apples comparisons of NAS approaches are difficult due to the existence of several hyperparameters influencing the final performance: various data augmentation techniques (including cutout [3] and auto-augment [4]), training epochs, learning rate decay methods ([5]), regularizing hyperparameters, label-smoothing, and others. Therefore, we have focused on comparing CoNAS with DARTs as a baseline.
>
> We agree that using 7 operations (8 operations, but with a zero operation) would enable a better one-to-one comparison to DARTs; however, we have observed from our experiments that the difference of operation sets between 5 ops and 7 ops is not a limiting factor.
>
> Table 1 shows that CoNAS outperforms not only with the experimental results for DARTs reported in [6] and [2], but also DARTs with 5 operations (6 operations, but zero operation). Moreover, the number of possible architectures ($\binom{140}{16} \approx 4.3 * 10^{20}$) from CoNAS far exceeds the search space of DARTs with 5ops ($5.0*10^{15}$, $7ops: 10^{18}$). Since CoNAS found the best architecture with less amount of search time even with the much larger search space, we infer that our improvements mainly arise due to the search method.
>
> Moreover, we would like to point out the performance of ENAS [7] taken from Table 1 from Liu et al. also uses 5 operations (6 ops including zero-ops), and achieve 2.91% test error with DARTs’ final training protocol, whereas CoNAS achieved 2.57% test error with significantly smaller parameter size (3.3M vs 2.3M).
>
>
> Q3: We thank the reviewer for raising this point. We have included the proof of Theorem 3.2 in the paper and some discussion in appendix A.2.
>
>
> Q4: We gratefully thank the reviewer for suggesting this structure of the paper. We have reorganized the paper accordingly to compress the main paper within 9 pages.
>
>
> Again, we appreciate your time and efforts looking into our paper.
>
>
> References
> [1] Gabriel Bender, Pieter-Jan Kindermans, Barret Zoph, Vijay Vasudevan, Quoc Le. Understanding and Simplifying One-Shot Architecture Search. In ICML, 2018
> [2] LIAM LI, AMEET TALWALKAR. Random Search and Reproducibility for Neural Architecture Search
> [3] Terrance DeVries, Graham W. Taylor. Improved Regularization of Convolutional Neural Networks with Cutout, 2017
> [4] Ekin D. Cubuk, Barret Zoph, Dandelion Mane, Vijay Vasudevan, Quoc V. Le. AutoAugment: Learning Augmentation Policies from Data, CVPR 2019
> [5] Andrew Hundt, Varun Jain, Gregory D. Hager. sharpDARTS: Faster and More Accurate Differentiable Architecture Search, 2019
> [6] Hanxiao Liu, Karen Simonyan, and Yiming Yang. DARTS: Differentiable architecture search. In ICLR, 2019.
> [7] Hieu Pham, Melody Y. Guan, Barret Zoph, Quoc V. Le, Jeff Dean. Efficient Neural Architecture Search via Parameter Sharing. In ICML, 2018

---

> > ### Comment · AnonReviewer4 · 2019-11-14
> > **Reviewer #4 rebuttal response**
> >
> > Thank you very much for answering to all the points I wrote on the review.
> >
> > However, I am not still convinced regarding the correlation between the stand-alone architectures evaluated with the one-shot weights vs. retrained from scratch.
> > Of course Bender et al. [1] show that this correlation is highly dependent on the max. drop path probability value used during the one-shot model train. But there where 2 recent papers [2, 3] that show that this correlation is actually low, and it a high value of that might be due to some intrinsic properties of the search spaces. Therefore, I believe that the method proposed might not scale to other space designs.
> >
> > Furthermore, I think that it is possible to compare apples-to-apples. There are certain benchmark papers doing this such as NAS-Bench-101 [4], NAS-Bench-1Shot1 [3], AA-NAS-Bench [5] or "NAS evaluation is frustratingly hard" [6]. Therefore I would say that running the proposed algorithm in the standard DARTS search space would be useful to assess the effectiveness of the proposed method alone, when compared to DARTS.
> >
> > I thank again the authors for their response and for taking into account the points I listed. However, I still would lean to a weak reject. I would recommend a more careful analysis to assess the sensitivity of their proposed method, and a more careful head-to-head comparison with other one-shot NAS algorithms using the same settings.
> >
> > -- References --
> > [1] Gabriel Bender, Pieter-Jan Kindermans, Barret Zoph, Vijay Vasudevan, Quoc Le. Understanding and Simplifying One-Shot Architecture Search. In ICML, 2018
> > [2] Christian Sciuto, Kaicheng Yu, Martin Jaggi, Claudiu Musat, Mathieu Salzmann. Evaluating the Search Phase of Neural Architecture Search. In ArXiv 2019.
> > [3] Anonymous submission at ICLR 2020. NAS-BENCH-1SHOT1: BENCHMARKING AND DISSECTING ONE-SHOT NEURAL ARCHITECTURE SEARCH
> > [4] Chris Ying, Aaron Klein, Esteban Real, Eric Christiansen, Kevin Murphy, Frank Hutter. NAS-Bench-101: Towards Reproducible Neural Architecture Search. In ICML 2019
> > [5] Anonymous submission at ICLR 2020. An Algorithm Agnostic NAS Benchmark.
> > [6] Anonymous submission at ICLR 2020. NAS evaluation is frustratingly hard.

---

### Decision · Program_Chairs · 2019-12-19

**Decision:**

Reject

**Comment:**

This paper proposed to use a compressive sensing approach for neural architecture search, similar to Harmonica for hyperparameter optimization.

In the discussion, the reviewers noted that the empirical evaluation is not comparing apples to apples; the authors could not provide a fair evaluation. Code availability is not mentioned. The proof of theorem 3.2 was missing in the original submission and was only provided during the rebuttal. All reviewers gave rejecting scores, and I also recommend rejection.